# Acetone Sensing and Catalytic Conversion by Pd-Loaded SnO_2_

**DOI:** 10.3390/ma14205921

**Published:** 2021-10-09

**Authors:** Pascal M. Gschwend, Florian M. Schenk, Alexander Gogos, Sotiris E. Pratsinis

**Affiliations:** 1Particle Technology Laboratory, Department of Mechanical and Process Engineering, Institute of Energy and Process Engineering, ETH Zurich, Sonneggstrasse 3, CH-8092 Zurich, Switzerland; gschwend@ptl.mavt.ethz.ch (P.M.G.); florian.schenk@ife.ee.ethz.ch (F.M.S.); 2Nanoparticle Systems Engineering Laboratory, Department of Mechanical and Process Engineering, Institute of Energy and Process Engineering, ETH Zurich, Sonneggstrasse 3, CH-8092 Zurich, Switzerland; alexander.gogos@empa.ch; 3Particles-Biology Interactions, Swiss Federal Laboratories for Materials Science and Technology (Empa), Department of Materials Meet Life, Lerchenfeldstrasse 5, CH-9014 St. Gallen, Switzerland

**Keywords:** chemoresistive, metal oxide, breath sensor, n-type, nanoparticles

## Abstract

Noble metal additives are widely used to improve the performance of metal oxide gas sensors, most prominently with palladium on tin oxide. Here, we photodeposit different quantities of Pd (0–3 mol%) onto nanostructured SnO_2_ and determine their effect on sensing acetone, a critical tracer of lipolysis by breath analysis. We focus on understanding the effect of operating temperature on acetone sensing performance (sensitivity and response/recovery times) and its relationship to catalytic oxidation of acetone through a packed bed of such Pd-loaded SnO_2_. The addition of Pd can either boost or deteriorate the sensing performance, depending on its loading and operating temperature. The sensor performance is optimal at Pd loadings of less than 0.2 mol% and operating temperatures of 200–262.5 °C, where acetone conversion is around 50%.

## 1. Introduction

Gas sensors have become ubiquitous in preventing gas explosions, poisoning, and drunk driving, as millions of sensors are sold each year [1]. Furthermore, they show increasing potential for medical diagnostics, especially as inexpensive and portable sensing devices become available [2]. In particular, chemoresistive metal oxide sensors are widespread due to their simplicity and low cost [3], among which the most prominent are made of SnO_2_, while Pd is the noble metal of choice as a dopant or additive [4]. 

In particular, the sensing mechanism of model gases such as CO or H_2_ have been studied thoroughly [5], focusing on reception (the interaction of reducing gases with the sensor surface), transduction (the resulting changes of the electronic properties of the sensing layer), and the influence of sensor morphology on sensing performance [6]. In particular, the role of noble metals has been investigated to boost sensor performance [7]. Specifically, the addition of Pd to SnO_2_ affects sensing through the spill-over mechanism (chemical sensitization) and electronic coupling (electronic sensitization). Still, even though the basic sensing principles are largely clear, the sensor design often follows an Edisonian approach.

In this regard, there are many similarities between gas sensing and heterogeneous catalysis. These two fields share similar preparation routes, characterization tools, and structure–function relations [8], yet few studies (Yamazoe et al. [7] for H_2_ and C_3_H_8_, Cabot et al. [9] for CH_4_, Ogel et al. [10] and Degler et al. [11] for CO) have considered both catalytic and sensing properties of chemoresistive materials. In particular, temperature effects on sensing are not always thoroughly investigated in sensor development. Often, the crystal size and composition as well as the noble metal loading are varied to optimize the performance of the sensor while operating it at a fixed temperature (typically between 300 and 400 °C) [12]. Only after this optimization, usually, the performance of the best sensor composition is evaluated at different operating temperatures, yielding the ideal one, typically close to the initial temperature [13].

The goal of this work is to systematically investigate Pd-loaded SnO_2_ sensors, focusing on their operating temperature, and explore synergies with the catalytic conversion of the analyte. Thus, acetone is important in breath analysis [14] and indoor air monitoring [15]. Specifically, breath acetone is a reliable marker [16] for ketogenic activity and a volatile byproduct [17] of lipolysis; therefore, it can serve to monitor the impact of diet and exercise on the human body. Here, the relationship between gas sensing and analyte conversion is investigated for better understanding and selection of the optimal sensing conditions and Pd loadings of the above sensors.

## 2. Materials and Methods

*Synthesis of Sensing Particles*: Pure SnO_2_ nanoparticles were prepared by flame spray pyrolysis [18]. As-prepared powders were annealed in air for five hours at 500 °C. Palladium was then photo-deposited [19] onto pure flame-made and annealed SnO_2_ at concentrations of 0, 0.1, 0.2, 0.5, 1, and 3 mol% using a Pd-nitrate solution. Afterwards, the washed and dried powders were annealed again at 500 °C for 5 h to further stabilize them [20].

*Powder characterization:* The particle composition was investigated by X-ray diffraction (XRD, Bruker D2 Phaser), while the specific surface area (SSA) of powders was determined by N_2_ adsorption. The noble metal dispersion was determined on an Autochem II (Micromeritics) by CO-pulse chemisorption.

For electron microscopy analysis, specimens were imaged by TEM (transmission electron microscopy, Talos F200X, Super-X EDS, 4 detector configuration, FEI, Hillsboro, OR, USA) or HAADF-STEM (high-angle annular dark-field scanning transmission electron microscopy) combined with EDX (energy-dispersive x-ray) elemental mapping.

For chemical analysis, the Pd-loaded particles were reduced, followed by leaching in HNO_3_ at 60 °C for 4 h. The leachate solution was analyzed by ICP-OES (inductively coupled plasma optical emission spectrometry, Varian 720-ES axial) to determine the Pd loading. 

*Sensor Fabrication:* The Pd-containing SnO_2_ particles were mixed with 1,2-propanediol (Aldrich, purity >99.5%) to form viscous and homogeneous pastes. Sensing films were prepared by doctor-blading [18] onto Al_2_O_3_ sensor substrates (15 × 13 × 0.8 mm, Electronic Design Centre Case Western Reserve University, Electrode type #103). The sensors were dried in ambient air for 4 h at 80 °C. No further annealing was done before sensing tests [21].

*Sensor measurements:* The sensors were tested as described previously [18]. In brief, 1 L/min of synthetic air with 50% relative humidity (RH) was admixed with acetone from calibrated gas standards. The sensor response (S) was defined as $S=R_{air}/R_{analyte}-1$, wherein *R_air_* and *R_analyte_* are the sensing film resistance in synthetic air (including RH) without and with analyte, respectively. The sensor response and recovery times were defined as the times needed to reach or recover 90% of the resistance change during or after analyte exposure, respectively.

*Catalytic measurements*: Catalytic measurements were performed as described previously [22] with a small modification: The catalytic packed beds consisted of only 20 mg particles in a quartz glass tube, which was placed inside a furnace, and 0.15 L/min of the same gases flowed through that tube, as for the sensing tests. The off-gas was analyzed in real-time by a PTR-ToF-MS (proton transfer reaction time of flight mass spectrometry, IONICON, PTR-ToF-MS 1000, Innsbruck, Austria) to its outlet.

## 3. Results and Discussion

### 3.1. Material Characterization

Figure 1a shows the XRD patterns of SnO_2_ with 0 and 3 mol% Pd (patterns for all Pd loadings are shown in Appendix A). They all exhibited the tetragonal cassiterite phase (squares). No peaks attributable to either Pd (circles) or PdO (triangles) were detected, even at the highest Pd loading, similar to FSP [18] and wet-made Pd-loaded SnO_2_. This can be attributed to small crystallite sizes, lack of crystallinity, and/or Pd content below the detection limit of XRD. The average SnO_2_ crystallite sizes (d_XRD_, inset Figure 1a and Appendix A) were between 18.6 nm (0 mol% Pd) and 16.8 nm (3 mol% Pd), indicating a slight inhibition of SnO_2_ growth by Pd during their annealing. The primary particle sizes, d_BET_ (inset table in Figure 1a and Appendix A), are larger (around 25 nm) than d_XRD_, most likely due to necking (Figure 1c). It is noteworthy that, after the first annealing step (before photodeposition), crystal (d_XRD_ = 16 nm) and especially primary particle (d_BET_ = 19.8 nm) sizes of pure SnO_2_ were slightly smaller, indicating that annealing increased necking in agreement with the relevant literature [18].

As a representative sample, the powder with 3 mol% Pd was investigated by microscopy and its particle size distribution (d_TEM_) is shown in Figure 1b. The particles exhibit a log-normal size distribution (fitted red line), typically observed for flame-made particles. The geometric mean (d_g_) was 15.3 nm, with a geometric standard deviation (σ_g_) of 1.34, thus slightly smaller than the crystal size (d_XRD_) and even smaller than the primary particle size (d_BET_). Interestingly, the second annealing step after photodeposition hardly influenced the d_TEM_ size distribution of SnO_2_ (Appendix A) similar to d_XRD_ (Figure 1a). The image (Figure 1c) also reveals the high crystallinity of particles, as well as some aggregation and sinter-necks, in line with the discussion above. No Pd particles could be identified by TEM, probably due to their small size or lack of contrast between Sn and Pd that have similar atomic numbers (Pd = 46, Sn = 50). Furthermore, the lattice fringes have been evaluated (Figure 1c, red arrows) with the most frequent spacing of 0.335 nm corresponding to the (1 1 0) plane of SnO_2_ [18]. Additionally, fringe spacings of 0.237 and 0.265 nm were observed (not shown), that were indexed to (2 0 0) and (1 0 1) of SnO_2_, respectively. It is noteworthy that the latter could also originate from the (1 1 1) of Pd, analogous to the XRD peak overlap at 34° (Figure 1a). In sum, Pd photodeposition hardly influenced the SnO_2_ support, which is characterized by partially aggregated particles of high crystallinity.

Having characterized the support, the Pd configuration was investigated. Specifically, its oxidation state, loading, dispersion, and cluster size are relevant, which depend on the precursor material [23], preparation technique [24], and subsequent treatment [20]. Here, UV-assisted photochemical deposition [25] was chosen for its high dispersion of Pd, where palladium ions are reduced to Pd metal clusters on the SnO_2_ surface. The deposited Pd affected the color of the powders, and their films became darker [26] for higher Pd loading (Figure 2a). 

After the first annealing, which has been reported to be crucial for stabilization [20], the appearance of the powders changed from black to brownish (Figure 2b), indicative of the formation of PdO_x_ [26]. Specifically, similar preparation conditions [27] led to formation of mostly Pd^2+^ (PdO) and some metal Pd, while a minor fraction (5%) Pd^3+^ was also detected that was presumably stabilized at the Pd cluster-SnO_2_ interface [28]. The actual Pd loadings determined by ICP-OES were close (7–14% relative deviation) to their nominal values (Table 1). To gather more information on Pd size, the 3 mol% Pd-loaded SnO_2_ sample was investigated using HAADF-STEM combined with EDXS (Figure 2). Using EDXS, it was possible to visualize the Pd clusters, which were otherwise indiscernible from Sn(O_2_) by bright-field TEM or HAADF-STEM (Appendix A). After identification by EDXS, an area with a predominant Pd signal was subsequently imaged with HR-TEM (Appendix A): no lattice fringes were detected for these Pd-spots or Pd-containing clusters, suggesting that such clusters were mostly amorphous PdO_x_.

Figure 2c shows an overlay of the elemental mappings of Sn and Pd of the annealed sample, while the as-prepared is shown in Appendix A: Pd is generally well-dispersed over the Sn signal, but differently sized Pd-containing clusters were observed and analyzed through their EDX spectra (Figure 2d–g). A large cluster with an area-equivalent circle diameter (d_eq_) of 11 nm diameter (Area 1, Figure 2d) showed the strongest peak in the EDX spectrum at 2.84 keV, corresponding to the Pd-L_α_ line. A medium-sized cluster of 3.6 nm (Area 2, Figure 2e) exhibited the same peak, but with lower intensity than the Sn-L_α_ line at 3.443 keV. To find the lower limit of cluster size that can still be identified with confidence, the analyzed cluster size was progressively decreased. At a size (d_eq_) of 2.8 nm (Area 3, Figure 2f), the Pd signal was low, but could still be unambiguously differentiated from bare SnO_2_ (Area 4, Figure 2g). Therefore, the cluster size was analyzed neglecting all clusters below 3 nm in area-equivalent (d_eq_) size. Still, it should be noted that this does not rule out the presence of smaller clusters. The geometric mean cluster size was 8.6 nm (Appendix A). Interestingly, before any annealing, more Pd clusters could be observed than afterwards (Appendix A), especially smaller ones, leading to a mean cluster size of only 4.6 nm for the as-prepared sample with 3 mol% Pd (Appendix A). This indicates some Pd cluster mobility and growth during annealing. Kutukov et al. [29] prepared similar nanostructures based on FSP-made SnO_2_ with 1 wt% Pd impregnated, followed by annealing at 300 °C. They observed two distinct fractions of small (<2 nm) and large (8–20 nm) Pd clusters.

While visualization by elemental mapping was possible for high Pd loadings, this became increasingly difficult for lower loadings. Elemental mappings of annealed SnO_2_ with 1 and 0.5 mol% Pd are shown in Appendix A: at 1 mol% Pd, still some Pd-clusters around 10 nm could be clearly distinguished (Appendix A) with a diffuse Pd-signal spread over the SnO_2_ particles. Reducing the Pd concentration down to 0.5 mol%, no more large Pd clusters could be identified (Appendix A), probably due to the highly homogeneous distribution of Pd as well as Pd-cluster sizes below 3 nm. Similarly, a diffuse Pd signal was observed at 3 mol% (Appendix A), yet it was not as visible due to the presence of larger clusters. Therefore, lowering the Pd loading reduces the Pd cluster sizes, while there is a fraction of small Pd clusters at all Pd loadings. 

In addition to elemental mappings, the CO-pulse chemisorption method was employed. The resulting dispersion values are given in Table 1, indicating the fraction of Pd available for CO-chemisorption. The dispersion values lie between 21.6 and 36.0%, whereas no clear trend with Pd loading can be observed. These values can also be converted into the corresponding Pd particle sizes given in Table 1: for 3 mol% Pd, the resulting size of 4.9 nm is below the observed mean value of 8.6 nm based on the electron microscopy analysis, which can be explained by the neglect of Pd clusters smaller than 3 nm in electron microscopy image counting (Appendix A, gray zones). Using the same precursors for Pd deposition with impregnation instead of photodeposition, Ma et al. [23] found a trend for increasing Pd size with higher Pd loading. Specifically, they determined a Pd size of 3.5 and 5.3 nm for 0.1 and 0.5 mol% Pd by CO chemisorption, respectively. In contrast, Takeguchi et al. [30] found a negligible influence of Pd loading (impregnation, up to 20 wt%) on the Pd cluster size. The use of photodeposition could influence the growth mechanism of the Pd clusters, as Pd ions have been observed to deposit preferentially on already existing Pd clusters [25]. In summary, XRD, BET, STEM-EDXS, and CO-chemisorption confirm the successful preparation of Pd-decorated SnO_2_ nanoparticles. These particles consist of Pd clusters that are in close contact and well-dispersed on Sn in an oxidized state, and between 3–6 nm mean Pd cluster sizes by CO-chemisorption for all Pd loadings. 

### 3.2. Sensing

The response to 1 ppm acetone at 50% relative humidity of all Pd-containing SnO_2_ sensors was evaluated as a function of temperature (Figure 3a). With increasing Pd loading, the maximum sensor response shifts to lower temperatures. For example, the sensor response of pure SnO_2_ peaks at 262.5 °C, while the sensor with 3% Pd loading exhibits its maximum response at 200 °C, the same as with 1 and 0.5% Pd. In terms of sensor response, all Pd-loaded samples outperform pure SnO_2_ at their respective ideal sensing temperature, reaching responses of up to 80 in the case of 1% Pd loading.

Such a shift to lower temperatures is consistent with Korotcenkov [21] who reported a strong shift in CO (0.5%) sensing from 421 °C in pure SnO_2_ to 153 °C for SnO_2_ with 1.1% Pd, with a corresponding increase in response by a factor of 2. Additionally, Zhang et al. [31] reported such a shift in sensing 100 ppm H_2_ from 320 °C without Pd to 280 °C at 3% Pd with doubling of the sensor response. Similarly, Yuasa et al. [25] observed a shift of the optimum between unloaded and 1 mol% Pd in H_2_ sensing from 350 to 300 °C and an increase in response by a factor of 2 to 200 ppm by increasing the Pd loading to 1%. Suematsu et al. [32] found a shift from 300 to 250 °C when increasing the Pd concentration from 0 to 0.2% in sensing 50 ppm of toluene along with an increased sensor response by a factor of 3. Lastly, Tang [33] reported a shift from 300 to 275 °C for 1.5% Pd compared to pure SnO_2_ in acetone (20 ppm) sensing and an increase in response again by a factor of 2. From the above, an optimal Pd loading for maximum sensor response is in the range of 0.1–1%. It is important to mention that the effect of Pd addition on the sensing performance is highly temperature-dependent (Figure 3a and Appendix A). For example, at 350 °C, any Pd deteriorates the acetone sensing performance, while at 250 °C, only 0.1% Pd outperforms the bare SnO_2_ sensor. At even lower temperatures (i.e., 200 °C), any addition of Pd drastically improves the acetone sensing performance (Figure 3a and Appendix A). 

The mechanism for acetone sensing proceeds as follows [34]: oxygen molecules chemisorb on the SnO_2_ surface and are ionized to oxygen ions through the capture of free electrons from SnO_2_, leading to an electron depletion layer. When that surface comes into contact with a reducing gas such as acetone, it reacts with the oxygen ions on the sensing film. This reaction releases trapped electrons back to the sensing material and reduces the electron depletion layer, resulting in a drop in resistance. 

Furthermore, Pd improves sensing through electronic sensitization [7], where PdO acts as an acceptor of electrons that are removed from the SnO_2_ surface to PdO. Thus, the electric resistance measurement in air (Appendix A) is an effective way to verify this effect [35]. Indeed, the baseline resistance increases with Pd loading and saturates at 0.5–1% Pd. This indicates a good electronic coupling between SnO_2_ and PdO_x_ clusters [25], and consequently, changes in the composition or stoichiometry will have an impact on the electrical properties of the sensing layer.

Besides sensitivity, the response and recovery times are crucial measures of sensor performance. The response times to 1 ppm acetone between 150 and 350 °C are shown in Figure 3b for all Pd loadings. Above 300 °C, the response is relatively fast, between 10 and 30 s for all Pd loadings. However, for lower temperatures, the response time increases up to 1000 s for pure SnO_2_ at 150 °C. The same trend was reported by Yin and Guo [36] for Pd/Fe co-loaded SnO_2_ to CO. Depending on the application, the requirements on the response time can be quite different. For example, in breath sensing, response times below 30 s are desired to reach a steady-state response within the duration of one buffered breath pulse [37]. This restricts already the use of operating temperatures below 212.5 °C.

Additionally, Figure 3c depicts the recovery time that increases at decreasing sensing temperatures similar to response time. In contrast to the response time, however, a clearer trend is observed for the recovery time as a function of Pd loading: at high temperatures (>225 °C), higher Pd-loadings shorten the recovery time, while at lower temperatures (<200 °C), this is reversed. This is likely caused by the interplay of temperature-dependent oxygen chemisorption, catalytic activity, and Pd oxidation. Still, even with high Pd loadings, the recovery time at 250 °C is 300 s already. While shorter recovery times are preferred, requirements are not as stringent as for the response times and also depend on application. For example, for monitoring ketogenic diets through breath acetone measurements, samples have been taken every 3 h [38]. However, if breath acetone measurements are employed as a tool for personalized fat burn monitoring, a higher frequency of pulses can be desirable (e.g., 45 min [39] down to 5 min [40] between pulses). As a result, in Figure 3c, the green-shaded area represents a recovery time of 3 min as a rough threshold for exercise monitoring, while the orange-shaded area represents a recovery time threshold of 30 min that can be acceptable for monitoring dieting. Still, the present response and recovery times could be further decreased by transient response analysis [41].

Based on the above, we selected the pure SnO_2_ sensor operating at 325 °C (t_response_ = 24 s, t_recovery_ = 104 s, response = 5.8) and the 0.1% Pd-loaded sensor operating at 237.5 °C (t_response_ = 26 s, t_recovery_ = 1361 s, response = 43.2) as the most promising acetone sensors requiring fast or highly sensitive sensing, respectively. The film resistance changes in response to ultralow acetone concentrations (20–50 ppb) are shown in Appendix A. The responses to 20 ppb could be easily distinguished with a signal-to-noise ratio (SNR) of 650 and 935 for the pure SnO_2_ (at 325 °C) and 0.1% Pd-loaded sensor (at 237.5 °C), respectively. While lower concentrations were not measured, the extrapolated limits of detection (LoD) defined at a SNR = 3 were approximately 0.05 ppb for both sensors. 

The performance of our sensors was compared to literature in terms of sensitivity, response/recovery times, and LoD for Pd-loaded SnO_2_ (Table 2) and other material compositions (Appendix A) acetone sensors. Appendix A details the precise origin of all literature entries of Table 2. The reported operating temperatures range from 250 to 400 °C, with one even at room temperature [34]. Looking at the sensor response, our sensors perform well, yet are outperformed in terms of response/recovery times when operated at lower temperatures. This can be attributed to slower diffusion and reaction kinetics. It should be noted that these times are not only influenced by Pd loading and temperature but also by film morphology (sometimes called “utility ratio” [42]). In summary, choosing the “best” sensor strongly depends on the targeted application and is always a compromise.

### 3.3. Catalytic Conversion of Acetone by Packed Beds of Pd-Loaded SnO_2_ Particles

The conversion of 1 ppm acetone in air at 50% relative humidity was investigated as a function of temperature by off-gas analysis with PTR-ToF-MS using packed beds of the sensing particles (Figure 4a). With bare SnO_2_, the conversion of acetone starts at 200 °C and reaches 100% at 300 °C. Increasing the Pd loading systematically shifts the acetone conversion to lower temperatures.

To quantitatively assess the catalytic activity, the activation energies were calculated based on the reaction rates at the low conversion range, as described in the Appendix A. The activation energies (Table 1) decrease with increasing Pd loading, from 119.7 kJ/mol for bare SnO_2_ to a minimum of 41.2 kJ/mol at 1% Pd. Comparable values have been obtained for catalytic combustion of acetone over Cu_0.13_Ce_0.87_O_y_ (97 kJ/mol) [43] and 3 wt% MnO_x_ on SiO_2_ (34 kJ/mol) [44]. Thus, the addition of Pd effectively lowers the activation barrier in acetone oxidation.

For further analysis, the temperature at 50% conversion of acetone by all Pd-loaded SnO_2_ packed beds is shown in Figure 4b (blue triangles) as a function of Pd loading. Already, the addition of small amounts of Pd strongly decreases the oxidation temperature that practically levels off at about 150 °C at 1% Pd. Similar behavior has been observed recently [50] in catalytic oxidation of acetone over Pt/Al_2_O_3_. However, a stronger temperature shift from 325 to almost 50 °C was observed, and leveling off started with a Pt loading of 3%. Such beneficial effects of Pd loading have been attributed to chemical sensitization [51]; palladium is a much better catalyst for activation of the dissociation of molecular oxygen than SnO_2_ [13]. Thus, PdO_x_ clusters form sites of dissociative adsorption, where reaction products (oxygen radicals) diffuse to the metal oxide support surface (spill-over effect) [52]. 

Figure 4b compares also the temperature for 50% acetone conversion (T_50%_) of Pd-loaded SnO_2_ (triangles) to their optimal sensing temperatures (circles). Both are quite similar, with a steep drop at low Pd loadings and leveling off at higher ones. Similarly, Yamazoe et al. [7] investigated both the catalytic and sensing properties of different metal-loaded SnO_2_ towards propane (0.2%) and hydrogen (0.8%), shown in Figure 4c as red squares and circles, respectively. They reported a correlation between ideal sensing temperature (in terms of maximum response) and T_50%_ conversion, independent of metal loading or analyte. By comparing our data (Figure 4c, blue triangles) with theirs, a similar trend is observed here for acetone. The shift of the present correlation to slightly lower conversion temperatures is probably due to different size (and/or porosity) of the catalytic bed and humidity of gas streams (0 [7] vs. 50%) or the much lower gas concentrations [53] (8000 ppm H_2_ and 2000 ppm propane vs. 1 ppm acetone).

## 4. Conclusions

Palladium-loaded SnO_2_ powders were prepared by photodeposition of Pd onto flame-made SnO_2_. The SnO_2_ particles were highly crystalline and not affected by the addition of Pd. Well-dispersed PdO_x_ clusters between 4 and 6 nm in diameter (and even finer) were revealed on the SnO_2_ surface. The optimal temperature for acetone sensitivity decreases with increasing Pd loading, while the sensor response and recovery times increase for lower temperatures. While the addition of Pd reduces these times, relatively high temperatures are still required for sensing of acetone for dieting guidance or exercise monitoring. The contribution of Pd to the sensing behavior was explained both by electronic and chemical sensitization. The complementary catalytic measurements revealed a strong correlation to the sensing temperature, where the highest responses were found close to 50% acetone conversion, representing a balance between signal reception and transduction. 

## Figures and Tables

**Figure 1 materials-14-05921-f001:**
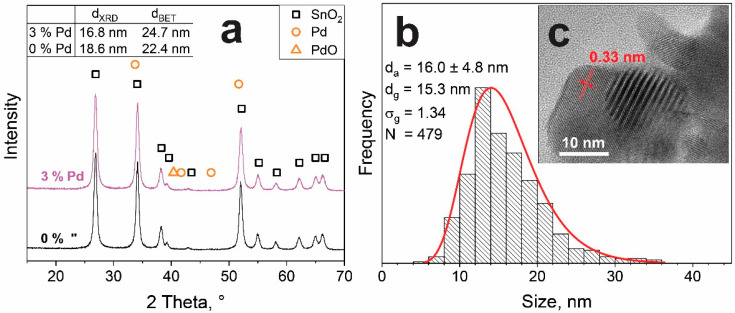
(**a**) XRD patterns of pure (0%) and 3 mol% Pd-loaded SnO_2_. Inset gives the crystal (d_XRD_) and primary particle sizes from N_2_-adsorption (d_BET_). (**b**) Primary particle size distribution of 3 mol% Pd-loaded SnO_2_ determined by electron microscopy and fitted with a log-normal curve (red line) with the arithmetic mean (d_a_), geometric mean (d_g_), geometric standard deviation (σ_g_), and number of counted particles (N). (**c**) Representative microscopy image of 3 mol% Pd-loaded SnO_2_. The (1 1 0) lattice spacing of SnO_2_ is indicated as 0.33 nm.

**Figure 2 materials-14-05921-f002:**
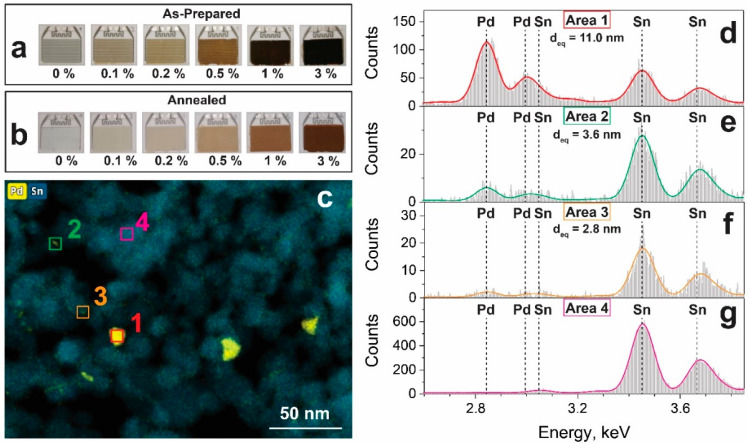
Sensor substrates covered with (**a**) as-prepared and (**b**) annealed films of Pd-loaded SnO_2_. (**c**) Overlayed elemental maps of Pd (yellow) and Sn (blue) of annealed 3 mol% Pd-loaded SnO_2_. The EDX spectra of selected areas are depicted in the right panel: a large Pd-containing cluster (Area 1), spectrum in (**d**), a medium-sized (Area 2), spectrum in (**e**) and a small one (Area 3), spectrum in (**f**) as well as a Pd-free SnO_2_ region (Area 4), spectrum in (**g**). Colored lines represent the fitted EDX peaks.

**Figure 3 materials-14-05921-f003:**
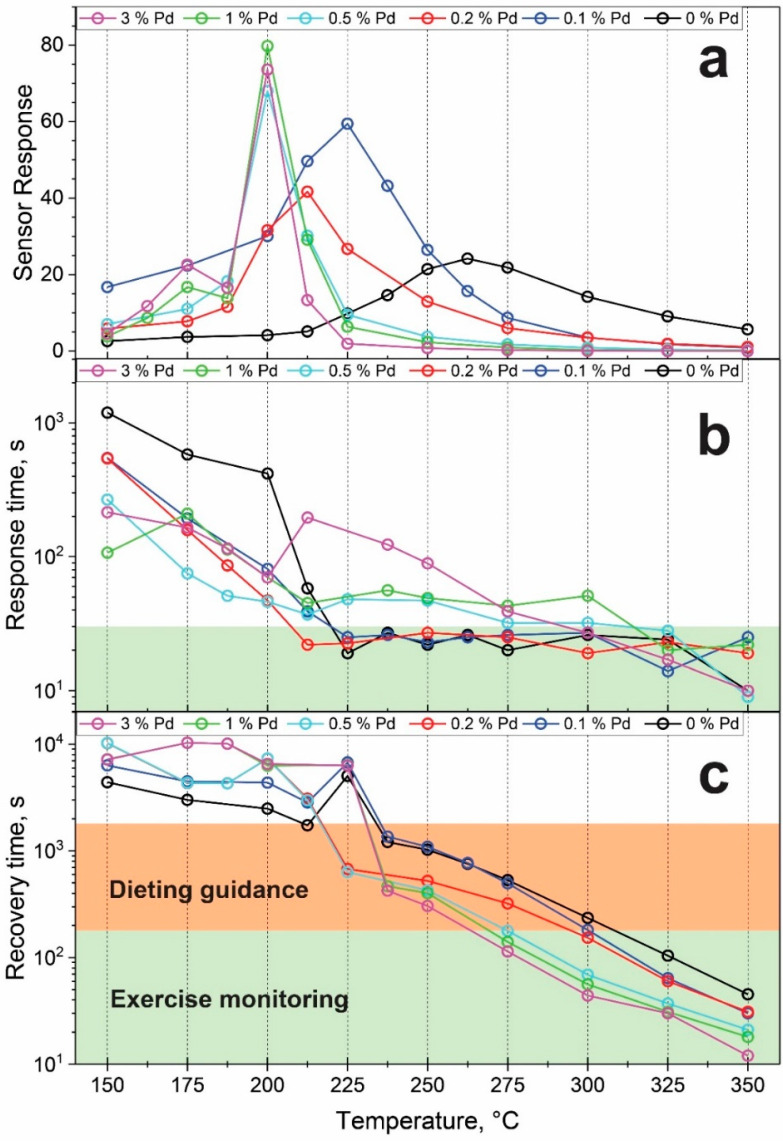
(**a**) Sensor response to 1 ppm of acetone (50% RH). Increasing the Pd loading shifts the sensing maximum to lower temperatures. (**b**) Response and (**c**) recovery times of these sensors.

**Figure 4 materials-14-05921-f004:**
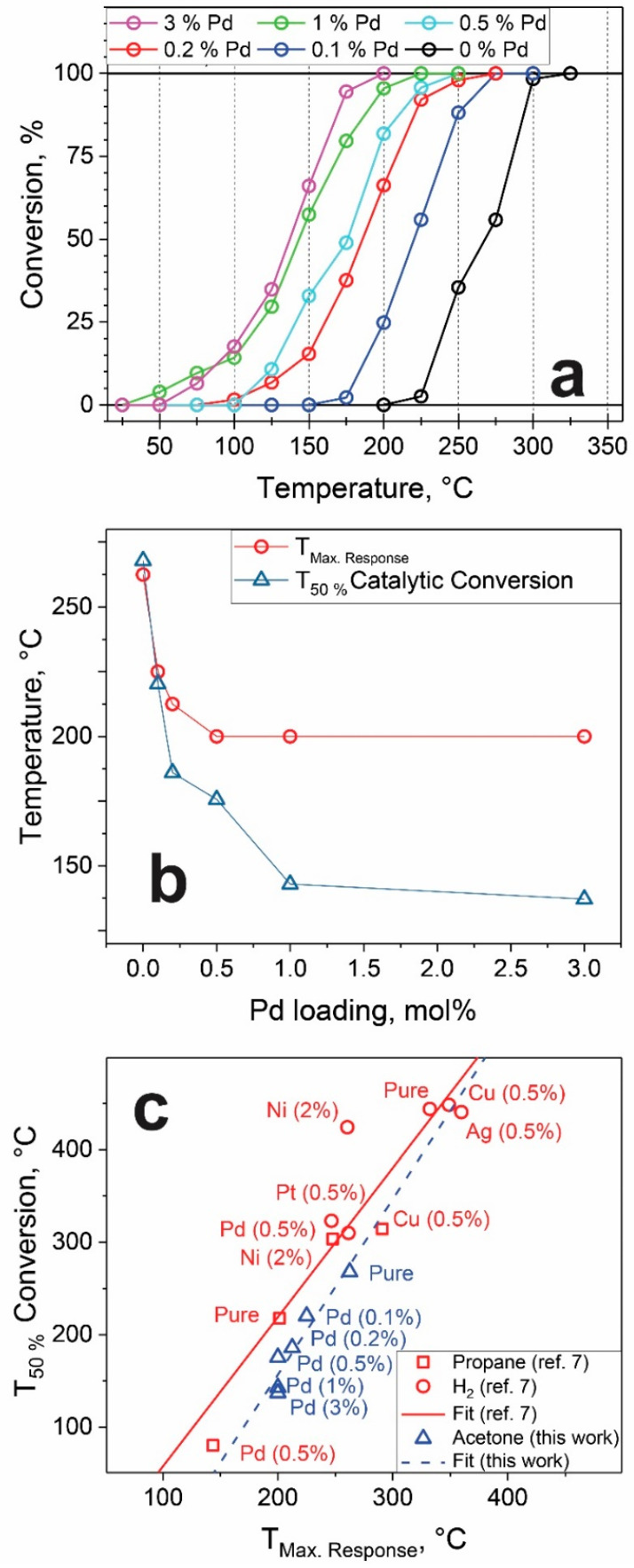
(**a**) Conversion of 1 ppm acetone; (**b**) Temperature of max. response and 50% catalytic conversion as a function of Pd loading; (**c**) Comparison of T_50%_ conversion and T_Max. Response_ by [7] and this work.

**Table 1 materials-14-05921-t001:** Palladium loading, dispersion, cluster size, and activation energy.

Nominal Loading (mol%)	Loading By ICP-OES (mol%)	Pd Dispersion ^a^ (%)	Pd Size ^a^ (nm)	Activation Energy ^b^ (kJ/mol)
0	-	-	-	119.7
0.1	0.11	21.6	5.2	101.8
0.2	0.18	24.4	4.6	64.1
0.5	0.54	35.8	3.1	53.8
1	1.07	36.0	3.1	41.2
3	2.60	22.8	4.9	51.5

^a^: Based on CO chemisorption. ^b^: Its determination is documented at the end of the Appendix A.

**Table 2 materials-14-05921-t002:** Comparison of reported Pd-SnO_2_ sensors for acetone detection with their key performance indicators.

Material	Operating Temp. [°C]	Relative Humidity [%]	Response ^a^ (Conc. in ppm)	Equiv. Response at 1 ppm ^b^	LOD [ppm]	Response Time ^c^ [s]	Recovery Time ^c^ [s]	Reference
Pd-SnO_2_	300 °C	Yes, but not specified	78 (25)	23.5	25 * 1 ^†^	n.a.	n.a.	Epifani et al. [45] (2008)
Pd-loaded flower-like SnO_2_	250 °C	No R.H.	10 (10)	1	10 *	11 (10 ppm)	30 (10 ppm)	Tian et al. [46] (2014)
Pd-SnO_2_ organized	RT	No R.H.	1.8 (10)	0.18	10 *	13	15	Shao et al. [34] (2015)
Pd-SnO_2_ nanofibers	275	No R.H.	3 (1)	3	1 *	20	40	Tang et al. [33] (2015)
Pd-loaded SnO_2_ ultrathin nanorod-assembled hollow microspheres	230	No R.H.	10.6 (20)	0.5	n.a.	n.a.	n.a.	Zhang et al. [47] (2017)
PdO@ZnO-SnO_2_ NT	400	95	4.1 (1)	4.1	0.1 * 0.01 ^†^	19.6	64	Koo et al. [48] (2017)
PdAu-SnO_2_ nanosheets	250	40–70%	2.7 (1)	2.7	0.1 * 0.045 ^†^	5	4	Li et al. [49] (2019)
Pd-doped SnO_2_	350	50	7 (1)	7	0.005 * 0.0005 ^†^	60 (50 ppb)	138 (50 ppb)	Pineau et al. [18] (2020)
SnO_2_	325	50	5.8 (1)	5.8	0.020 * 0.00005 ^†^	24	104	This work
0.1% Pd-loaded SnO_2_	237.5	50	43.2 (1)	43.2	0.020 * 0.00005 ^†^	26	1361	This work

a: Responses were converted to the definition used in this manuscript. b: If not available, response was linearly extrapolated. c: Concentration was 1 ppm if not stated otherwise. †: LOD calculated/estimated. *: Lowest measured concentration.

## Data Availability

Data are contained within the article or Appendix A.

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
