# Peer review of "Acetone Sensing and Catalytic Conversion by Pd-Loaded SnO2"

_materials, 2021, doi:10.3390/ma14205921_

Round 1

Reviewer 1 Report

Comment 1: In the last line of abstract you should mention the temperature.

Comment 2: Line 21 of introduction - This is confusing. In this line, shouldn't you mention compositions instead of temperatures? Please check!

Comment 3: Page 3, Line 15 - Pd-nitrate solution, or powder?

Comment 4: Page 6, Lines 6-7 - Please insert a picture showing these values of dXRD and dBET.

Comment 5: Page 9, Line 12 - In Table 1 you have 21.6 and 36.0%.

Comment 6: Page 14, Line 28 - This is only for Tmax, because T50% seems lower!

Comment 7: Figure 4b - Replace by "Temperature of 50% catalytic conversion and Tmax response as a function of Pd loading.

Comment 8: Line 4 of Conclusions - The sensor response only increases with Pd loading at lower temperatures, the same for recovery times!

Comment 9: Figure S8 - You say the scale bars are 50 nm, but is it for 1 cm?? It is not visible!

Reviewer 2 Report

In this manuscript, the authors photodeposit different quantities of Pd onto nanostructured SnO2 and determine their effect on sensing acetone, a critical tracer of lipolysis by breath analysis. Before the acceptation, there are some modification need to be done.

1. Please explain in the “introduction” why acetone is selected as the detected gas of Pd-loaded SnO2 sensors, what is the significance of this catalytic system on it.
2. In Figure 3C, why the trend before and after 200-225℃ is so different, the author can add discussion and analysis here.

Reviewer 3 Report

The authors present the development of a SnO with Pd sensor for acetone and the catalytic conversion by SnO2 loaded with Pd. The topic is of relevant scientific interest, for the topic "Advanced materials for gas sensors" however, making a very general and superficial review of the manuscript, several aspects can be observed, which must be corrected or improved.

- The manuscript is completely without the format of the magazine and it is not pleasant or easy to review an article without the specific format, it distracts a lot from the main content of the topic in question.

- It is also very difficult to make particular comments without having the lines numbered.

- It is suggested to improve the abstract.

- The keywords should be changed and improved to be more consistent with the content of the work and the topic of the special edition.

- The introduction must be revised and improved and errors such as the reference number format must be corrected.

- The experimental part is too explained and developed and it is not common to find such a descriptive article, it is suggested to summarize this section. Also, it is not common to find websites in the reagent specifications, it is suggested to delete.

- The conclusions are very brief, although concrete and summarized, it is suggested to highlight the characterization results by relating them to the results of the tests of both the sensor and those of the catalytic activity.

- The references, they are not cited with the format of the journal, nor are they indicated in the text in the correct way and it is difficult to locate them.

- I suggest that the supplementary material be presented in a separate file because it is very difficult to consult it having to go through the entire manuscript.

Reviewer 4 Report

The manuscript "Acetone Sensing and Catalytic Conversion by Pd-loaded SnO2", authors P. M. Gschwend et al. deals with the study of Pd-SnO2 as sensors for acetone focusing on the role of the Pd loading and the operating temperature on the sensor response and recovery time. Interestingly, a correlation between the efficiency of the prepared materials as acetone sensors and for the catalytic conversion of acetone was carried out. The manuscript is well written, the morphological and structural properties of the prepared materials are well investigated by different spectroscopic and microscopy techniques, as well as the catalytic and sensor properties are extensively studied and discussed. For these reasons, the contribution can be accepted in the present form for the publication in "Materials" journal.  

Reviewer 5 Report

Revision of “Acetone Sensing and Catalytic Conversion by Pd-loaded SnO2

The manuscript under review devoted to the systematically investigate Pd-loaded SnO2 sensors focusing on their operating temperature and explore synergies with the catalytic conversion of the analyte. Providing of such investigations is very important from an academic point of view (giving new knowledge about the nature of the objects under study) and economic (new high sensitivity sensors).

Authors rather thoroughly described the method of obtaining experimental samples in Experimental Section. So, palladium-loaded SnO2 powders were prepared by photodeposition of Pd onto flame-made SnO2. The SnO2 particles were highly crystalline and not affected by the addition of Pd. It was found that the optimal temperature for acetone sensitivity decreases with increasing Pd loading while the sensor response and recovery times increase. While the addition of Pd reduces these times, still relatively high temperatures are required for dieting guidance (250 °C) or exercise monitoring (325 °C). The complementary catalytic measurements revealed a strong correlation to the sensing temperature, where the highest responses were found close to 50 % acetone conversion, representing a balance between signal reception and transduction.

In manuscript all necessary information is captured by 16 figures and 3 table. There are 86 references, all of them are adequate and are reflected in the text.

After getting acquainted with the presented manuscript, a small imperfection:

Point 1. The submitted work must be completed according to the requirements of the Journal.

The obtained results are important both for understanding the physical processes that occur in real objects and for the development of new materials. The described manuscript is sufficient, comprehensive and it corresponds to the field of the Journal «Materials». However, I would recommend the publication of the presented material in the journal «Sensors».

Round 2

Reviewer 5 Report

Revision of “Acetone Sensing and Catalytic Conversion by Pd-loaded SnO2

The manuscript under review devoted to the systematically investigate Pd-loaded SnO2 sensors focusing on their operating temperature and explore synergies with the catalytic conversion of the analyte. Providing of such investigations is very important from an academic point of view (giving new knowledge about the nature of the objects under study) and economic (new high sensitivity sensors).

Authors rather thoroughly described the method of obtaining experimental samples in Experimental Section. So, palladium-loaded SnO2 powders were prepared by photodeposition of Pd onto flame-made SnO2. The SnO2 particles were highly crystalline and not affected by the addition of Pd. It was found that the optimal temperature for acetone sensitivity decreases with increasing Pd loading while the sensor response and recovery times increase. While the addition of Pd reduces these times, still relatively high temperatures are required for dieting guidance (250 °C) or exercise monitoring (325 °C). The complementary catalytic measurements revealed a strong correlation to the sensing temperature, where the highest responses were found close to 50 % acetone conversion, representing a balance between signal reception and transduction.

The obtained results are important both for understanding the physical processes that occur in real objects and for the development of new materials. The described manuscript is sufficient, comprehensive and it corresponds to the field of the Journal «Materials». However, I would recommend the publication of the presented material in the journal «Sensors».